# Learning Robust Anymodal Segmentor with Unimodal and Cross-modal Distillation

## Abstract

Leveraging multimodal inputs from multiple sensors offers intuitive benefits for semantic segmentation but introduces practical challenges—most notably, unimodal bias, where models overfit to dominant modalities and perform poorly when others are missing, a common issue in real-world scenarios. To address this, we propose *AnySeg*, a unified framework for learning robust segmentors that generalize to arbitrary combinations of input modalities. Our approach first trains a strong multimodal teacher using parallel modality learning. We then distill both unimodal and cross-modal knowledge to an anymodal student via multiscale feature-level distillation, reducing modality dependence and improving generalization. To further enhance semantic consistency, we introduce a prediction-level, modality-agnostic distillation loss. Unlike prior work, our framework explicitly handles missing modalities challenges by learning ***unimodal and cross-modal*** correspondence among input modalities. Extensive experiments on synthetic and real-world multi-sensor datasets demonstrate the effectiveness of AnySeg, achieving notable improvements of **+6.37%** and **+6.15%** in mIoU.

## 1 Introduction

The success of multimodal deep learning relies heavily on effectively leveraging information from multiple modalities, particularly for complex tasks such as semantic segmentation in scene understanding (Zheng et al., 2024d; Lyu et al., 2024a; Zheng et al., 2024a; Lyu et al., 2024b). While intuitively beneficial, training segmentation models, *a.k.a.*, segmentors, with inputs from multiple sensors presents significant practical challenges. A prominent issue in this context is **unimodal bias** — a phenomenon where networks develop an over-reliance on a single, faster-to-learn modality, often overlooking other sources of valuable information. This bias stems from the distinct characteristics and varied learning dynamics of each sensor modality.

For example, the well-known CMX model (Zhang et al., 2023a) in multimodal semantic segmentation suffers significant performance drops when evaluated without the RGB modality. Meanwhile, the state-of-the-art model Any2Seg (Zheng et al., 2024b), which aims at learning modality-agnostic representation for missing modality problems, demonstrates a significant performance decline when evaluated in modality-incomplete scenarios. For instance, when depth data is missing, segmentation performance drops markedly (RD: 68.21 → R: 39.02, a ***decrease*** of ***29.19*** mIoU), illustrating how unimodal bias can lead to substantial performance degradation in real-world applications where certain modalities are often unavailable.

Despite advancements in multimodal learning, such as leveraging large multimodal language models (Zheng et al., 2024b) and prioritizing each modality (Zheng et al., 2024c), progress in addressing unimodal bias and fostering robust multimodal correlations remains limited. To address this gap, we introduce the first framework for learning robust anymodal segmentors[1]. This framework is tailored to handle real-world scenarios where modality completeness cannot be guaranteed, such as missing modality (Liu et al., 2024) or modality-agnostic segmentation (Zheng et al., 2024b).

Our approach begins with a novel **P**arallel **M**ultimodal **L**earning (**PML**) strategy, which facilitates the learning of a strong teacher model for both unimodal and multimodal distillation *without adding extra parameters*. Inspired by recent methods (Zheng et al., 2024b;c), we process all multimodal inputs

---

[1]We define anymodal segmentors as models that ensure robust performance despite missing modalities.

from different sensors in a single mini-batch, passing them through the segmentation backbone, *i.e.*, SegFormer. Multimodal fusion is performed through simple averaging, and supervision is applied at the final layer of the segmentation decoder. This straightforward yet effective PML strategy enables segmentor to focus on capturing both unimodal and multimodal knowledge (See Tab. 5 and Tab. 6).

We then introduce a dual-level distillation process: Unimodal Distillation (UMD) and Cross-modal Distillation (CMD), applied across multi-scale representations and prediction levels. To simulate real-world scenarios, we apply an anymodal dropout strategy, where the multimodal inputs are randomly masked, creating varied modality combinations within each batch. For distribution distillation within the multi-scale representation space, the features from the anymodal segmentor are trained to align with the corresponding features from the multimodal teacher, thereby replicating the unimodal feature extraction capabilities. Furthermore, cross-modal correspondence is applied across all active modalities to mitigate the effects of unimodal bias. Finally, at the prediction level, we employ modality-agnostic semantic distillation to facilitate effective task-specific knowledge transfer between teacher and student models, further enhancing the robustness in diverse real-world conditions.

Extensive experiments on real-world and synthetic benchmarks demonstrate the superior robustness and performance of our method compared to existing state-of-the-art approaches, achieving mIoU improvements of **+6.37%** and **+6.15%**, respectively. Moreover, we analyze why fused multimodal fusion distillation is **unsuitable** for ensuring robustness in multimodal segmentation and further discuss the feature characteristics of multimodal data.

## 2 RELATED WORK

**Multimodal Semantic Segmentation** Semantic segmentation with multi-sensor inputs enhances scene understanding by leveraging complementary information from diverse sensors, such as event cameras (Zhou et al., 2024; Zheng & Wang, 2024), LiDAR sensors (Li et al., 2023), and others (Liao et al., 2025b;a; Zheng et al., 2023; 2024e;d). Recent advances in multi-sensor systems have led to the development of various approaches (Zheng et al., 2025; 2024c;b; Zhang et al., 2023a) and datasets (Zhang et al., 2023b; Brödermann et al., 2024) that extend from dual-modality fusion to full multimodal fusion, with the aim of achieving robust perception across diverse lighting and environmental conditions throughout the day (Zhao et al., 2025; Broedermann et al., 2023; Wei et al., 2023; Zhang et al., 2021; Man et al., 2023; Wang et al., 2022; Chen et al., 2021; Zhang et al., 2023b;a; Zhu et al., 2024). For instance, MUSES (Brödermann et al., 2024) dataset integrates data from a frame camera, LiDAR, radar, event camera, and IMU/GNSS sensors to capture driving scenes in adverse conditions with increased uncertainty. Recently, CMNeXt (Zhang et al., 2023b) introduced the task of fusing an arbitrary number of modalities, although this approach still relies primarily on RGB input for optimal performance. In our work, we address the challenge of ***unimodal bias*** in multimodal semantic segmentation by focusing on developing a robust anymodal segmentor that can maintain performance across various input combinations, rather than solely optimizing for multimodal segmentation accuracy.

**Missing Modality Robustness** In the multimodal learning community, several studies have sought to understand unimodal bias from both empirical (Kleinman et al., 2023; Peng et al., 2022) and theoretical perspectives (Huang et al., 2022). As shown by Huang *et. al* (Huang et al., 2022; Zhang et al., 2024), while multimodal learning has the potential to surpass unimodal performance, it often falls short due to modality competition: only the subset of modalities more closely aligned with the encoder's initial parameters tends to dominate learning within the multimodal network. This phenomenon also occurs in multimodal semantic segmentation, as MAGIC (Zheng et al., 2024c) and Any2Seg (Zheng et al., 2024b) struggle when depth data is missing during inference. In this work, we focus on addressing practical challenges in multi-sensor systems that are widely applicable across industrial domains, including autonomous driving and intelligent systems. We define the unimodal bias problems in multimodal semantic segmentation and propose the anymodal semantic segmentation framework.

**Robust Multimodal Segmentors.** In practice, sensor failures often result in incomplete multimodal data, challenging segmentation frameworks typically trained on complete modality pairs (Liu et al., 2024). Recent studies aim to build models that, while trained with full modalities, remain effective when some inputs are missing (Liu et al., 2024; Wang et al., 2023b; Maheshwari et al., 2024; Reza et al., 2023; Chen et al., 2023; Zhao et al., 2023). Wang *et. al* (Wang et al., 2023a) proposed

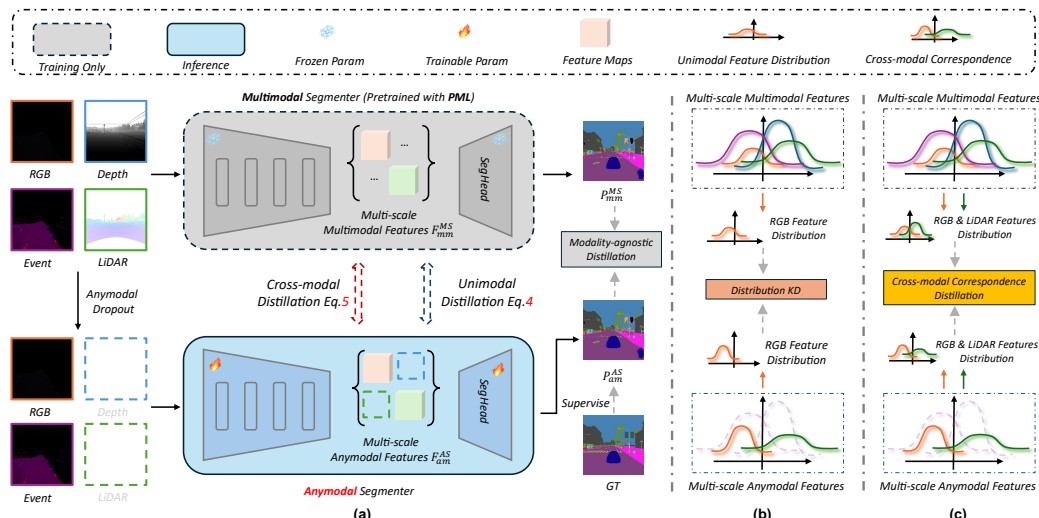

Figure 1: (a) Overall of AnySeg with a two-stage training strategy: the multimodal teacher is first trained using PML, then frozen for student distillation. (b) Unimodal feature distillation transfers intra-modality knowledge. (c) Cross-modal feature distillation enables modality interaction transfer.

adaptive modality selection and knowledge distillation for cross-modal compensation. Liu *et. al* (Liu et al., 2024) extended this to modality-incomplete scene segmentation, addressing both system- and sensor-level failures. More recent methods like MAGIC (Zheng et al., 2024c) and Any2Seg (Zheng et al., 2024b) aim for modality-agnostic segmentation by extracting shared representations. However, unimodal bias remains unresolved. For instance, Any2Seg's performance drops sharply without depth input (RD: 68.21 → R: 39.02), illustrating the challenge. To address this, we propose the *first* framework for learning robust anymodal segmentors capable of handling missing modalities by distilling both unimodal and cross-modal knowledge. We also introduce a parallel multimodal learning strategy to build a strong teacher model, further advancing robust segmentation under incomplete inputs.

## 3 METHODOLOGY

The overall framework is depicted in Fig. 1. It consists of two segmentors, *i.e.*, the multimodal teacher segmentor $\mathcal{F}_{ms}$ and anymodal student segmentor $\mathcal{F}_{as}$, as well as two key modules, including the unimodal and cross-modal distillation and the modality-agnostic semantic distillation modules. The teacher $\mathcal{F}_{ms}$ is first pre-trained with our proposed parallel multimodal learning strategy to learn a strong teacher with expertise in multimodal scenarios, its parameter is frozen during training the student $\mathcal{F}_{as}$. **Inputs:** Our framework processes multi-modal visual data from four modalities, all within the same scene. We consider RGB images $\mathbf{R} \in \mathbb{R}^{h \times w \times 3}$, depth maps $\mathbf{D} \in \mathbb{D}^{h \times w \times C^D}$, LiDAR data $\mathbf{L} \in \mathbb{L}^{h \times w \times C^P}$, and event stack images $\mathbf{E} \in \mathbb{E}^{h \times w \times C^E}$ to illustrate our method, as depicted in Fig. 1. Here, we follow the data processing as (Zhang et al., 2023b), where the channel dimensions $C^D = C^P = C^E = 3$, and we also integrates the corresponding ground truth $Y$ across $K$ categories. For each training iteration, a mini-batch $\{r, d, e, l\}$ contains samples from all the input modalities.

### 3.1 PARALLEL MULTIMODAL LEARNING (PML) STRATEGY

Recent studies have shown that treating all input modalities equally can enhance both multimodal and unimodal performance (Zheng et al., 2024b;c). Building on insights from MAGIC (Zheng et al., 2024c), we adopt a uniform approach for handling all multimodal inputs and introduce a parallel multimodal learning strategy to train a robust teacher model for knowledge distillation. As illustrated in Fig. 2, we compute the mean across multimodal features at each block of the segmentation backbone (Xie et al., 2021). This averaged output serves as the input for the segmentation head, leading to improved multimodal performance, particularly on real-world benchmarks, achieving

51.37 mIoU on MUSES. The supervised loss $L_{pre}$ for training is:

$$\mathcal{L}_{pre} = -\sum_{i=1}^{N}\sum_{k=1}^{K} y_{i,k}\log(p_{i,k}), \tag{1}$$

where $N = h \times w$ is the total number of pixels, $y_{i,k}$ is the ground truth label for class $k$ at pixel $i$, and $p_{i,k}$ is the predicted probability for class $k$ at pixel $i$. $\mathcal{L}_{pre}$ encourages accurate predictions across all modalities and contributes to the robustness of the teacher model.

## 3.2 UNIMODAL AND CROSS-MODAL DISTILLATION

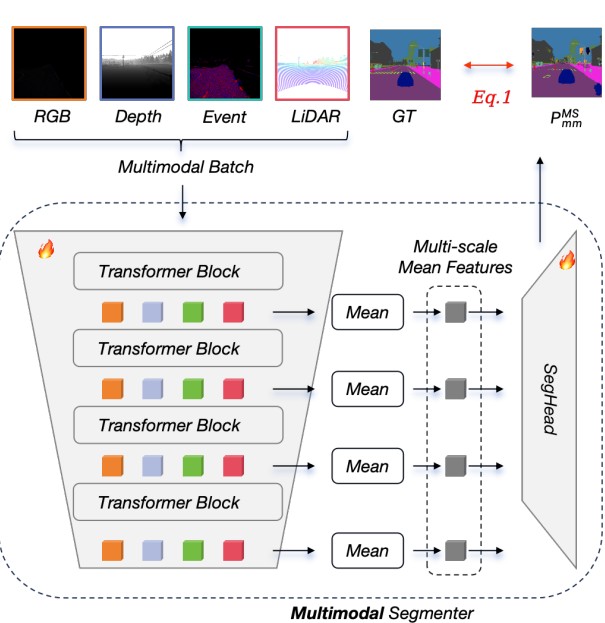

Figure 2: PML for learning a strong multimodal segmentor as teacher model.

After obtaining the strong multimodal teacher $\mathcal{F}_{ms}$, we turn to learn efficient and robust anymodal segmentor $\mathcal{F}_{as}$. As depicted in Fig. 1, the input multimodal mini-batch $\{r, d, l, e\}$ is directly ingested by weight-shared encoder within the multimodal segmentor $\mathcal{F}_{ms}$. This process yields multi-scale features $\{f_r^i, f_d^i, f_e^i, f_l^i\}_{i=1}^4$:

$$\{f_r^i, f_d^i, f_e^i, f_l^i\}_{i=1}^4 = F_{ms}(\{r, d, l, e\}), \tag{2}$$

where $i$ represents the multi-scale feature level, summed from 1 to 4. Meanwhile, the input multimodal mini-batch $r, d, l, e$ undergoes random masking to generate an anymodal batch, where modality data is randomly dropped from the batch, ensuring that at least one modality is retained in each instance. The anymodal batch is then processed by the encoder of the anymodal segmentor, yielding features $g_r^i, g_d^i{}_{i=1}^4$ [2] for the retained modalities:

$$\{g_r^i, g_d^i\}_{i=1}^4 = \mathcal{F}_{as}(\{r, d\}). \tag{3}$$

This process ensures that the anymodal segmentor is trained on diverse input combinations, improving its robustness and adaptability to incomplete data.

**Unimodal Distillation.** After extracting the multi-scale features $\{f_r^i, f_d^i, f_e^i, f_l^i\}_{i=1}^4$ and $\{g_r^i, g_d^i\}_{i=1}^4$ from the multimodal segmentor (teacher model) and the anymodal segmentor (student model), respectively, we proceed with the distillation process. For the remaining multi-scale features $\{g_r^i, g_d^i\}_{i=1}^4$ from the anymodal segmentor $\mathcal{F}_{am}$, we align them with the corresponding features $\{f_r^i, f_d^i, f_e^i, f_l^i\}_{i=1}^4$ obtained from the multimodal segmentor.

The unimodal knowledge distillation loss function based on KL divergence is defined as:

$$\mathcal{L}_{umd} = \sum_{i=1}^{4}\left(\sum_{j=1}^{C_i} \tilde{g}_r^{i,j}\log\left(\frac{\tilde{g}_r^{i,j}}{\tilde{f}_r^{i,j}}\right) + \sum_{j=1}^{C_i} \tilde{g}_d^{i,j}\log\left(\frac{\tilde{g}_d^{i,j}}{\tilde{f}_d^{i,j}}\right)\right), \tag{4}$$

where $C_i$ denotes the number of channels in the $i$-th level features. To ensure valid probability distributions for KL divergence computation, we apply *softmax* to the teacher features $g$ and *log-softmax* to the student features $f$, yielding normalized representations $\tilde{g}$ and $\tilde{f}$. This guarantees ***non-negativity and unit sum***, thus satisfying the theoretical prerequisites of KL divergence. The loss term $\mathcal{L}_{umd}$ promotes intra-modality knowledge transfer, enhancing the anymodal segmentor's ability to generalize from unimodal inputs. This is demonstrated in Tab. 5, where we show the performance improvements achieved by applying unimodal distillation in the anymodal segmentation task. However, while unimodal knowledge transfer enhances single-modality performance, especially

---

[2]For example, we illustrate the case where the event and LiDAR modalities are dropped.

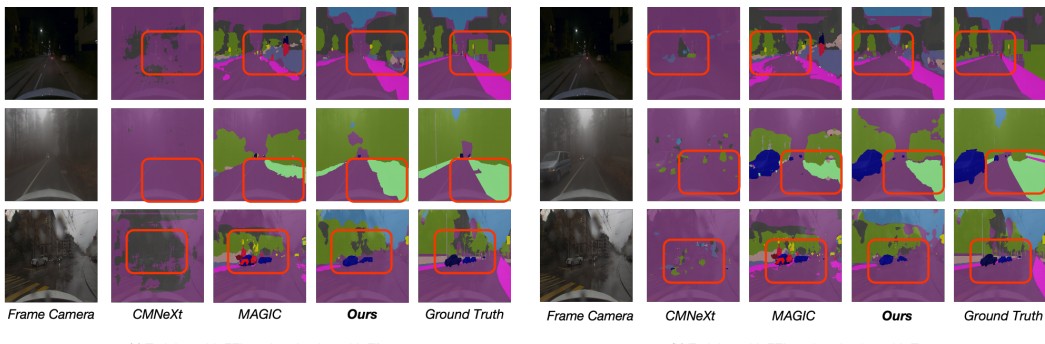

Frame Camera  CMNeXt  MAGIC  **Ours**  Ground Truth    Frame Camera  CMNeXt  MAGIC  **Ours**  Ground Truth

(a) Training with FEL and evaluation with **EL**            (b) Training with FEL and evaluation with **E**

Figure 3: Qualitative comparison on MUSES.

for the RGB images, it simultaneously hinders multimodal performance when different modality combinations are encountered, as also illustrated in Tab. 5.

**Cross-modal Correspondence Distillation.**

While unimodal knowledge distillation significantly improves segmentation performance on RGB images, as shown in Tab. 5, it also introduces unimodal bias that reduces performance on other modalities. This bias causes the model to over-rely on the RGB modality, which is easier for the model to learn, thereby limiting its generalization capacity across diverse input types. Consequently, unimodal knowledge transfer, while beneficial for single-modality performance, negatively impacts multimodal performance in mixed-modality scenarios. This is evident in Tab. 5, where performance decreases for modalities such as Event (-3.74% ↓) and LiDAR (-2.91% ↓).

To address this, we leverage cross-modal correspondences between "easy-to-learn" and "hard-to-learn" modalities to achieve a more balanced performance. By distilling these cross-modal relationships from the teacher to the student model—referred to as the "anymodal" segmentor—we effectively mitigate unimodal bias across all modalities.

The distillation of cross-modal knowledge between student features $\{g_r^i, g_d^i\}_{i=1}^4$ and teacher features $\{f_r^i, f_d^i, f_e^i, f_l^i\}_{i=1}^4$ is achieved through:

$$\mathcal{L}_{cmd} = \sum_{i=1}^{4} \sum_{j=1}^{C_i} \tilde{\mathcal{S}}\left(g_d^{i,j}, g_r^{i,j}\right) \log \left( \frac{\tilde{\mathcal{S}}\left(g_d^{i,j}, g_r^{i,j}\right)}{\tilde{\mathcal{S}}\left(f_d^{i,j}, f_r^{i,j}\right)} \right), \tag{5}$$

where $\mathcal{S}(x,y) = \frac{x \cdot y}{\|x\|\|y\|}$ denotes the cosine similarity between feature vectors $x$ and $y$, and $\tilde{\mathcal{S}}(x,y) = \frac{\mathcal{S}(x,y)+1}{2}$ is its **normalized form within** $[0,1]$ to ensure numerical stability for the logarithmic operation. To further mitigate potential instability caused by negative cosine values—especially when the similarities of teacher and student features differ in sign—we *average the similarity scores across batch samples with the same feature shape*. Empirically, this averaging leads to *non-negative values* when modalities are semantically aligned. Nonetheless, we adopt the normalized similarity $\tilde{\mathcal{S}}$ to guarantee theoretical robustness. This formulation aligns cross-modal representations by minimizing the discrepancy between teacher and student similarities across modalities. After applying cross-modal correspondence distillation, the inherent unimodal bias in this task—as well as the bias introduced by unimodal knowledge distillation—are largely mitigated, as shown in Tab. 5.

### 3.3 MODALITY-AGNOSTIC DISTILLATION

After addressing the unimodal bias problem in the representation spaces, we also focus on transferring task-related semantic information at the prediction level for further utilization of the pre-trained knowledge in the teacher model. Specifically, the segmentation maps predicted by the multimodal teacher $P_{mm}$ are used as supervision signals for the predictions of the anymodal student segmentor $P_{am}$. The modality-agnostic distillation loss is formulated as:

$$\mathcal{L}_{mad} = \frac{1}{N} \sum_{i=1}^{N} \sum_{k=1}^{K} P_{am}^{i,k} \log \left( \frac{P_{am}^{i,k}}{P_{mm}^{i,k}} \right). \tag{6}$$

Table 1: Results of anymodal semantic segmentation validation with three modalities (F: frame camera, E: event cameras, L: LiDAR sensor) on real-world MUSES with SegFormer-B0.

| Method | Pub. | Training | Anymodal Evaluation | | | | | | | Mean |
|---|---|---|---|---|---|---|---|---|---|---|
| | | | F | E | L | FE | FL | EL | FEL | |
| CMX (Zhang et al., 2023a) | T-ITS 2023 | | 2.52 | 2.35 | 3.01 | 41.15 | 41.25 | 2.56 | 42.27 | 19.30 |
| CMNeXt (Zhang et al., 2023b) | CVPR 2023 | | 3.50 | 2.77 | 2.64 | 6.63 | 10.28 | 3.14 | 46.66 | 10.80 |
| MAGIC (Zheng et al., 2024c) | ECCV 2024 | FEL | 43.22 | 2.68 | _22.95_ | 43.51 | 49.05 | _22.98_ | 49.02 | 33.34 |
| Any2Seg (Zheng et al., 2024b) | ECCV 2024 | | _44.40_ | _3.17_ | 22.33 | 44.51 | _49.96_ | 22.63 | _50.00_ | _33.86_ |
| Ours | - | | **46.01** | **19.57** | **32.13** | **46.29** | **51.25** | **35.21** | **51.14** | **40.23** |
| _w.r.t_ SoTA | - | - | +1.61 | +16.40 | +9.80 | +1.78 | +1.29 | +12.58 | +1.14 | +6.37 |

Table 2: Results of anymodal semantic segmentation validation with four modalities (R: RGB, D: Depth, E: Event, L: LiDAR) on DELIVER using SegFormer-B0 as backbone.

| Method | Anymodal Evaluation | | | | | | | | | | | | | | Mean |
|---|---|---|---|---|---|---|---|---|---|---|---|---|---|---|---|
| | R | D | E | L | RD | RE | RL | DE | DL | EL | RDE | RDL | REL | DEL | RDEL | |
| CMNeXt (Zhang et al., 2023b) | 0.86 | 0.49 | 0.66 | 0.37 | 47.06 | 9.97 | 13.75 | 2.63 | 1.73 | _2.85_ | 59.03 | 59.18 | 14.73 | _59.18_ | 39.07 | 20.77 |
| MAGIC (Zheng et al., 2024c) | 32.60 | _55.06_ | 0.52 | _0.39_ | _63.32_ | 33.02 | 33.12 | _55.16_ | _55.17_ | 0.26 | _63.37_ | _63.36_ | _33.32_ | 55.26 | _63.40_ | 40.49 |
| Any2Seg (Zheng et al., 2024b) | _39.02_ | **60.11** | _2.07_ | 0.31 | **68.21** | _39.11_ | _39.04_ | **60.92** | **60.15** | 1.99 | **68.24** | **68.22** | 39.06 | **60.95** | **68.25** | 45.04 |
| Ours | **47.11** | 52.17 | **17.33** | **19.01** | 60.37 | **47.49** | **48.13** | 52.82 | 52.29 | **21.47** | 60.16 | 60.60 | **47.98** | 52.44 | 60.26 | **46.64** |
| _w.r.t_ SoTA | **+14.51** | -2.89 | **+16.81** | **+18.62** | -2.95 | **+14.47** | **+15.01** | -2.34 | -2.88 | **+21.21** | -3.21 | -2.76 | **+14.66** | -2.82 | -3.14 | **+6.15** |

Table 3: Ablation study of different loss combinations on MUSES dataset (Brödermann et al., 2024).

| Loss Combination | F | $\Delta\uparrow$ | E | $\Delta\uparrow$ | L | $\Delta\uparrow$ | FE | $\Delta\uparrow$ | FL | $\Delta\uparrow$ | EL | $\Delta\uparrow$ | FEL | $\Delta\uparrow$ | Mean | $\Delta\uparrow$ |
|---|---|---|---|---|---|---|---|---|---|---|---|---|---|---|---|---|
| $\mathcal{L}_{sup}$ | 43.69 | - | 22.35 | - | 32.14 | - | 44.58 | - | 48.53 | - | 35.40 | - | 48.35 | - | 39.29 | - |
| $\mathcal{L}_{sup} + \lambda\mathcal{L}_{mad}$ | 43.71 | +0.02 | 23.00 | +0.65 | 34.70 | +2.56 | 44.18 | -0.40 | 49.13 | +0.60 | 37.23 | +1.83 | 48.79 | +0.44 | 40.11 | +0.82 |
| $\mathcal{L}_{sup} + \lambda\mathcal{L}_{mad} + \alpha\mathcal{L}_{umd}$ | 45.82 | +2.13 | 19.26 | -3.09 | 31.79 | -0.35 | 45.88 | +1.30 | 51.11 | +2.58 | 33.56 | -1.84 | 50.60 | +0.43 | 39.72 | +0.43 |
| $\mathcal{L}_{sup} + \lambda\mathcal{L}_{mad} + \alpha\mathcal{L}_{umd} + \beta\mathcal{L}_{cmd}$ | 46.01 | +2.32 | 19.57 | -2.78 | 32.13 | -0.01 | 46.29 | +1.71 | 51.25 | +2.72 | 35.21 | -0.21 | 51.14 | +2.79 | 40.23 | +0.94 |

Table 4: Ablation study on the effect of different parameters b $L_{mad}$ in our framework on MUSES dataset (Brödermann et al., 2024). More results can be found in Table 11.

| $\lambda$ | F | $\Delta\uparrow$ | E | $\Delta\uparrow$ | L | $\Delta\uparrow$ | FE | $\Delta\uparrow$ | FL | $\Delta\uparrow$ | EL | $\Delta\uparrow$ | FEL | $\Delta\uparrow$ | Mean | $\Delta\uparrow$ |
|---|---|---|---|---|---|---|---|---|---|---|---|---|---|---|---|---|
| 1 | 43.97 | - | 22.33 | - | 31.90 | - | 44.82 | - | 48.61 | - | 35.14 | - | 48.33 | - | 39.30 | - |
| 20 | 44.08 | +0.11 | 22.76 | +0.43 | 32.35 | +0.45 | 44.37 | -0.45 | 49.33 | +0.72 | 34.73 | -0.41 | 48.79 | +0.46 | 39.49 | +0.19 |
| **50** | 43.71 | -0.26 | 23.00 | +0.67 | **34.70** | +2.80 | 44.18 | -0.64 | 49.13 | +0.52 | **37.23** | +2.09 | **48.79** | +0.46 | **40.11** | **+0.81** |
| 80 | 43.84 | -0.13 | 22.86 | +0.53 | 33.78 | +1.88 | 44.25 | -0.57 | 49.43 | +0.82 | 36.57 | +1.43 | 48.72 | +0.39 | 39.92 | +0.62 |

Additionally, there is also a supervised loss imposed between the $P_{am}$ and the GT:

$$\mathcal{L}_{sup} = -\frac{1}{N}\sum_{i=1}^{N}\sum_{k=1}^{K} y_k^i \log\left(P_{am}^{i,k}\right). \tag{7}$$

The total loss for training the anymodal student segmentor combines the supervised objective with multiple distillation terms, formulated as:

$$\mathcal{L}_{\text{total}} = \mathcal{L}_{sup} + \lambda_{\text{mad}}\mathcal{L}_{mad} + \alpha\mathcal{L}_{umd} + \beta\mathcal{L}_{cmd}, \tag{8}$$

where $\lambda_{\text{mad}}$, $\alpha$, and $\beta$ are weighting coefficients for the modality-adaptive, unimodal, and cross-modal distillation losses, respectively. The supervised loss $\mathcal{L}_{sup}$ serves as the primary training signal and is left unweighted for clarity, with all distillation losses scaled relative to it.

## 4 EXPERIMENTS

**Experimental Setup.** We evaluate our method on synthetic and real-world multi-sensor datasets. The MUSES dataset (Brödermann et al., 2024), recorded in Switzerland, includes driving sequences designed to address challenges from adverse visual conditions. It features multi-sensor data from a

Table 5: Ablation on different parameters for $L_{umd}$ on MUSES dataset (Brödermann et al., 2024).

| $\alpha$ | F | Δ↑ | E | Δ↑ | L | Δ↑ | FE | Δ↑ | FL | Δ↑ | EL | Δ↑ | FEL | Δ↑ | Mean | Δ↑ |
|---|---|---|---|---|---|---|---|---|---|---|---|---|---|---|---|---|
| w/o | 43.71 | - | 23.00 | - | 34.70 | - | 44.18 | - | 49.13 | - | 37.23 | - | 48.79 | - | 40.11 | - |
| 3 | 45.38 | +1.67 | 20.64 | -2.36 | 31.37 | -3.33 | 45.43 | +1.25 | 50.53 | +1.40 | 33.65 | -3.58 | 49.93 | +1.14 | 39.56 | -0.55 |
| **5** | 45.82 | +2.11 | 19.26 | -3.74 | 31.79 | -2.91 | 45.88 | +1.70 | 51.11 | +1.98 | 33.56 | -3.67 | 50.60 | +1.81 | 39.72 | -0.39 |
| 7 | 46.09 | +2.38 | 17.84 | -5.16 | 31.81 | -2.89 | 46.18 | +2.00 | 51.36 | +2.23 | 33.43 | -3.80 | 51.01 | +2.22 | 39.67 | -0.44 |
| 10 | 46.17 | +2.46 | 15.74 | -7.26 | 31.95 | -2.75 | 46.37 | +2.19 | 51.17 | +2.04 | 33.26 | -3.97 | 51.08 | +2.29 | 39.39 | -0.72 |

Table 6: Ablation on different parameters for add $L_{cmd}$ with $L_{umd}$ on MUSES (Brödermann et al., 2024). w/o means the framework is only trained with $L_{mad} + L_{umd}$. More results in Table. 13.

| $\beta$ | F | Δ↑ | E | Δ↑ | L | Δ↑ | FE | Δ↑ | FL | Δ↑ | EL | Δ↑ | FEL | Δ↑ | Mean | Δ↑ |
|---|---|---|---|---|---|---|---|---|---|---|---|---|---|---|---|---|
| w/o | 45.82 | - | 19.26 | - | 31.79 | - | 45.88 | - | 51.11 | - | 33.56 | - | 50.60 | - | 39.72 | - |
| 1 | 45.93 | +0.11 | 18.76 | -0.50 | 31.84 | +0.05 | 45.96 | +0.08 | 51.22 | +0.11 | 33.49 | -0.07 | 50.82 | +0.22 | 39.72 | 0.00 |
| 3 | 46.04 | +0.22 | 17.74 | -1.52 | 31.42 | -0.37 | 46.08 | +0.20 | 51.27 | +0.16 | 33.46 | -0.10 | 50.99 | +0.39 | 39.57 | -0.15 |
| 5 | 46.19 | +0.37 | 17.27 | -1.99 | 31.03 | -0.76 | 46.27 | +0.39 | 51.34 | +0.33 | 33.40 | -0.16 | 51.05 | +0.45 | 39.51 | -0.21 |
| 7 | 46.21 | +0.39 | 17.40 | -1.86 | 31.06 | -0.73 | 46.37 | +0.49 | 51.29 | +0.18 | 33.79 | +0.23 | 51.12 | -0.12 | 39.60 | -0.12 |
| **10** | 46.01 | +0.19 | 19.57 | +0.31 | 32.13 | +0.34 | 46.29 | +0.41 | 51.25 | +0.14 | 35.21 | +1.65 | 51.14 | +0.54 | 40.23 | +0.51 |
| 13 | 46.57 | +0.75 | 18.24 | -1.02 | 30.88 | -0.91 | 46.74 | +0.86 | 51.09 | -0.02 | 33.88 | +0.32 | 50.76 | +0.16 | 39.74 | +0.02 |
| 15 | 46.03 | +0.21 | 14.10 | -8.90 | 31.12 | -3.58 | 45.99 | +1.81 | 50.97 | +1.84 | 31.42 | -5.81 | 50.49 | -0.11 | 38.59 | -1.13 |
| 20 | 45.95 | +0.13 | 15.19 | -7.81 | 30.61 | -4.09 | 45.80 | +1.62 | 51.19 | +2.06 | 30.55 | -6.68 | 50.41 | +1.62 | 38.53 | -1.58 |

high-resolution frame camera, an event camera, and MEMS LiDAR, enhancing annotation quality and supporting robust multimodal semantic segmentation. Each sequence is annotated with 2D panoptic labels for accurate ground truth. The DELIVER dataset (Zhang et al., 2023b) includes RGB, depth, LiDAR, and event data across 25 semantic categories, covering various environmental conditions and sensor failures for thorough evaluations. We follow the official data processing and split protocols. More implementation details can be found in Sec. A.2

**Results** As shown in Tab. 1, our method achieves the highest mIoU of **40.23**, surpassing all state-of-the-art (SoTA) baselines. CMX (Zhang et al., 2023a) and CMNeXt (Zhang et al., 2023b) perform poorly (mIoU: 19.30% and 10.80%) due to over-reliance on RGB. Similarly, MAGIC (Zheng et al., 2024c) and Any2Seg (Zheng et al., 2024b) heavily depend on depth, leading to significant performance drops in its absence—highlighting their unimodal bias. In contrast, our method shows balanced performance across all modalities: RGB (46.01%), Event (19.57%), and LiDAR (32.13%), as well as paired combinations: FE (46.29%), FL (51.25%), EL (35.21%), and FEL (51.14%), demonstrating robust cross-modal learning. Significant gains in Event (+16.40%) and FL (+12.58%) further emphasize our method's ability to handle sparse or challenging modalities. Fig. 3 provides qualitative comparisons. Tab. 2 reports results on the synthetic DELIVER benchmark using SegFormer-B0. Our method achieves a top mIoU of **46.64%**, outperforming MAGIC (Zheng et al., 2024c) by **+6.15%**. For individual modalities, it achieves 47.11% (R), 52.17% (D), and 19.01% (L), with gains of +14.51% (R) and +18.62% (L) over MAGIC. Event and LiDAR improvements confirm robustness to unimodal bias. For paired modalities, RD, RE, and RL achieve mIoUs of 60.37%, 47.49%, and 48.13%, with notable gains over MAGIC (+14.47% RE, +15.01% RL). These results demonstrate our model's ability to capture cross-modal dependencies while remaining resilient to missing inputs. Overall, the results confirm the effectiveness of our method in both unimodal and multimodal settings, offering strong generalization across diverse sensor combinations.

## 5 ABLATION STUDY

**Effectiveness of Loss Functions** Tab. 3 reports results of different loss combinations on the MUSES dataset (Brödermann et al., 2024). Using only supervised loss ($\mathcal{L}_{sup}$) yields a mean mIoU of 39.29%. Adding $\lambda\mathcal{L}_{mad}$ improves performance across paired and combined modalities. Further including $\alpha\mathcal{L}_{umd}$ brings notable gains, especially in F (+2.13%), FE (+1.30%), and FL (+2.58%), confirming the benefit of unimodal knowledge distillation. The full combination with $\beta\mathcal{L}_{cmd}$ achieves the best performance (mean mIoU **40.23%**), with broad improvements, particularly in FL (+2.72%) and FEL (+2.79%). Even E improves slightly (+0.43%), showing robustness across modalities. In sum,

Table 8: Discussion study on the effect of performing KD with fused features on MUSES dataset (Brödermann et al., 2024). w/o means the framework is only trained with $L_{mad} + L_{umd} + L_{cmd}$.

| $\lambda$ | F | $\Delta\uparrow$ | E | $\Delta\uparrow$ | L | $\Delta\uparrow$ | FE | $\Delta\uparrow$ | FL | $\Delta\uparrow$ | EL | $\Delta\uparrow$ | FEL | $\Delta\uparrow$ | Mean | $\Delta\uparrow$ |
|---|---|---|---|---|---|---|---|---|---|---|---|---|---|---|---|---|
| w/o (ours) | 46.01 | - | 19.57 | - | 32.13 | - | 46.29 | - | 51.25 | - | 35.21 | - | 51.14 | - | 40.23 | - |
| 1 | 46.32 | +0.31 | 17.99 | -1.58 | 31.38 | -0.75 | 46.80 | +0.51 | 51.01 | -0.24 | 33.92 | -1.29 | 50.95 | -0.19 | 39.77 | -0.46 |
| 3 | 46.34 | +0.33 | 17.27 | -2.30 | 31.36 | -0.77 | 46.81 | +0.52 | 51.20 | -0.05 | 33.87 | -1.34 | 51.13 | -0.01 | 39.71 | -0.52 |

each loss contributes to better segmentation, and the full loss design is most effective for capturing complex multimodal interactions.

**Ablation on Teacher Model and Fusion Strategy** We compare our PML with fusion-based teacher models MAGIC and CMNeXt to evaluate their impact on student performance. Unlike CMNeXt's fixed fusion architecture, PML supports both unimodal and cross-modal distillation via parallel modality learning, offering more diverse and semantically aligned supervision.

Results in Tab. 7 show that distilling from fused features, as in MAGIC and CMNeXt, degrades performance. CMNeXt yields the largest drop (–4.34% mIoU), indicating that rigid fusion limits generalization. In contrast, PML achieves the best performance by preserving both modality-specific and shared representations. These results confirm PML's flexibility and effectiveness as a general-purpose teacher for anymodal segmentation.

Table 7: Ablation on teacher model selection.

| Teacher | F | E | L | FE | FL | EL | FEL | Mean |
|---|---|---|---|---|---|---|---|---|
| MAGIC | 43.87 | 13.22 | 30.90 | 43.91 | 48.41 | 33.68 | 47.75 | 37.39 (-2.84) |
| CMNeXt | 43.80 | 10.79 | 23.15 | 48.34 | 43.97 | 33.52 | 47.64 | 35.89 (-4.34) |
| PML (Ours) | 46.01 | 19.57 | 32.13 | 46.29 | 51.25 | 35.21 | 51.14 | **40.23** |

**Ablation on Hyper-Parameter Selection** We study the effect of hyper-parameters $\lambda$, $\alpha$, and $\beta$ for $\mathcal{L}_{mad}$, $\mathcal{L}_{umd}$, and $\mathcal{L}_{cmd}$, respectively (Tab. 4–6). Increasing $\lambda$ improves performance up to a point, particularly benefiting underrepresented modalities, before plateauing due to diminishing returns. Varying $\alpha$ reveals that moderate values enhance unimodal contributions, while overly large values lead to degradation. Adding $\mathcal{L}_{cmd}$ with an appropriate $\beta$ further improves results, but excessive weighting can suppress unimodal learning.

**Rationality of Unimodal Distillation** The effectiveness of $\mathcal{L}_{umd}$ is demonstrated in Tab. 5. Adding $\mathcal{L}_{umd}$ with $\alpha = 1$ improves single-modality performance, particularly for RGB ($F : +0.83\%$) and paired modalities such as FL ($+0.42\%$). The best performance for single-modality tasks is observed at $\alpha = 10$, where the mean mIoU for RGB increases to **46.17** ($+2.46\%$), and FL achieves **51.17%** ($+2.19\%$). Obviously, $\mathcal{L}_{umd}$ effectively improves single-modality segmentation performance by facilitating knowledge transfer within each modality. However, the results also highlight a trade-off: while unimodal distillation enhances performance for individual modalities, it may compromise the model's ability to handle complex multimodal combinations. Careful tuning of $\alpha$ is therefore critical to achieving a balance between single-modality and multimodal segmentation performance.

**Rationality of Cross-modal Distillation.** Unimodal knowledge distillation improves segmentation performance on RGB images but introduces unimodal bias, reducing performance on other modalities. This bias arises as the model over-relies on RGB, which is easier to learn, limiting its generalization across diverse inputs. As shown in Tab. 5, performance on Event (-3.74%) and LiDAR (-2.91%) modalities declines significantly. While unimodal knowledge transfer enhances single-modality performance, it negatively impacts multimodal performance in mixed-modality scenarios. Cross-modal correspondence distillation mitigates this bias by aligning representations across modalities, improving segmentation performance on diverse inputs. It balances the trade-off between single-modality and multimodal performance, emphasizing the need to carefully tune $\beta$ for optimal results. However, larger $\beta$ values ($\beta > 13$) degrade performance, particularly on Event (-8.90%) and LiDAR (-3.58%), suggesting that overemphasizing cross-modal distillation can overshadow individual modality learning, leading to performance trade-offs.

**Why not doing KD between Fused Features?** To explore transferring knowledge from teacher to student models via fused features, we conduct experiments on the MUSES dataset, using the fusion method described in (Zheng et al., 2024c). The results, summarized in Tab. 8, show that applying knowledge distillation (KD) directly on fused features leads to a significant performance drop across

Table 9: Performance under different conditions in the DELIVER with RGB-D modalities.

| Metric | All | Cloud | Fog | Night | Rain | Sun | M.B. | O.E. | U.E. | L.J. | E.L. |
|--------|-----|-------|-----|-------|------|-----|------|------|------|------|------|
| mIoU (%) | 55.99 | 57.94 | 55.02 | 54.64 | 56.24 | 56.82 | 54.73 | 53.71 | 53.15 | 54.66 | 54.18 |
| mF1 (%) | 66.97 | 68.80 | 65.33 | 66.16 | 67.43 | 67.46 | 64.88 | 64.59 | 64.10 | 64.51 | 65.04 |
| mAcc (%) | 64.58 | 65.83 | 64.51 | 63.62 | 64.91 | 64.41 | 63.82 | 61.88 | 62.06 | 62.46 | 62.65 |

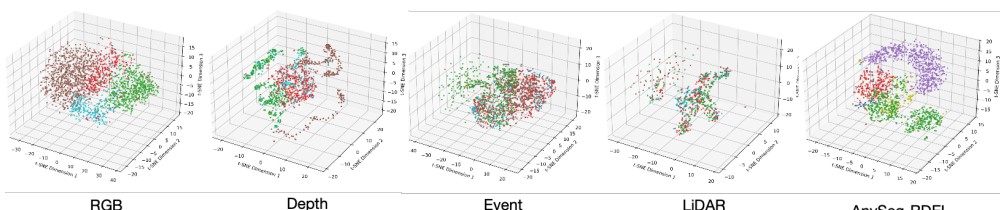

| RGB | Depth | Event | LiDAR | AnySeg-RDEL |

Figure 4: TSNE visualization of multi-modal features (RGB-R, Depth-D, Event-E, and LiDAR-L) and the learned features of our AnySeg framework.

most metrics. For example, compared to the baseline (without KD on fused features), which achieves a mean mIoU of 40.23%, all KD settings show reduced performance. At $\lambda = 1$, the mean mIoU drops to 39.77% (-0.46%), and higher $\lambda$ values worsen the decline, with $\lambda = 10$ yielding a mean mIoU of 39.37% (-0.86%). Examining individual modalities shows a consistent trend: for Event (E) and LiDAR (L), performance degrades as $\lambda$ increases, with Event mIoU dropping by -3.96% at $\lambda = 10$. Paired and fused modalities show minimal or negative improvement, such as FEL, which gains +0.03% at $\lambda = 5$ but regresses at $\lambda = 10$. These results suggest that distilling knowledge directly from fused features introduces noise and misalignment, limiting their effectiveness. This highlights the need for specialized distillation mechanisms targeting individual or structured features, rather than indiscriminately fused representations.

**t-SNE Visualization** Fig. 4 presents t-SNE visualizations of multimodal features, including RGB, Depth, Event, LiDAR, and the features under our AnySeg framework. Individual modality plots reveal distinct clusters, reflecting semantic separability. RGB and Depth exhibit relatively compact clusters, indicating strong discriminative power, whereas Event and LiDAR show more dispersed and overlapping clusters, highlighting weaker performance when used independently. The learned feature spaces, particularly AnySeg-RDL and AnySeg-RDEL, show notable improvements in cluster compactness and separability. AnySeg-RDEL, integrating RGB, Depth, Event, and LiDAR, achieves the most coherent and well-separated clusters, demonstrating the framework's robustness in leveraging complementary information across modalities. These results underscore AnySeg's effectiveness in addressing the limitations of individual modalities, achieving robust multi-modal feature representation, and enhancing segmentation performance through cross-modal fusion.

**Cross-Sensor Generalization under Scene Degradations.** We test AnySeg's performance on the DELIVER using RGB-D under various challenging conditions. As shown in Tab. 9, AnySeg maintains consistent performance across diverse sensor and environmental degradations. The model handles cloud, rain, and sun conditions well, with only minor drops under more severe challenges like motion blur, extreme exposure, and sensor noise (e.g., lidar jitter, event resolution). Despite these disruptions, the model achieves over 53% mIoU in all cases, demonstrating strong generalization.

## 6 CONCLUSION

In this paper, we addressed the challenge of unimodal bias in multimodal semantic segmentation, where reliance on specific modalities leads to performance drops when modalities are missing. We proposed the first framework for anymodal segmentation using unimodal and cross-modal distillation. A PML strategy ensures a strong teacher model, while multiscale distillation transfers feature-level knowledge. By distilling unimodal distributions with cross-modal correspondences, we reduce modality dependency. Additionally, modality-agnostic semantic distillation enables robust prediction-level knowledge transfer. Experiments on synthetic and real multi-sensor benchmarks validate the superior performance of our framework. Furthermore, our discussion on fused feature distillation highlights the need for specialized mechanisms targeting individual or structured features rather than indiscriminate fusion.

## REPRODUCIBILITY STATEMENT

The project code and experimental results on the MUSE dataset are available in the *Supplementary Materials*. All relevant resources, including implementation details and experimental data, are either available or have been made available to ensure the reproducibility of our work.

## ETHICS STATEMENT

This work does not involve human participants, personal data, or sensitive content, and it does not raise specific ethical concerns. Therefore, no additional ethical statements are required.

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

# A    APPENDIX

## A.1    LIMITATIONS & BROADER IMPACT

A key limitation of AnySeg is the additional computational cost introduced by the teacher-student framework for improving missing modality robustness and performance during training. This work advances multi-modal machine learning, particularly in visual pattern recognition, by addressing unimodal bias in multimodal segmentors. These models often struggle when certain modalities are missing, a common challenge in real-world applications. While our approach has potential societal impacts, we do not identify any specific concerns at this time.

## A.2    IMPLEMENTATION DETAILS.

All experiments on MUSES were conducted on 8 NVIDIA 3090 GPUs, while experiments on DELIVER utilized 4 NVIDIA A100 GPUs. The initial learning rate was set to $6 \times 10^{-5}$ and adjusted using a polynomial decay strategy with a power of 0.9 over 200 epochs. Additionally, a 10-epoch warm-up phase was applied at 10% of the initial learning rate to stabilize training. The AdamW optimizer was employed, and the batch size was set to 16. Input modality data was cropped to $1024 \times 1024$ resolution for consistency across benchmarks.

## A.3    QUALITATIVE FEATURE VISUALIZATION

We provide additional qualitative comparisons that highlight our method's robustness under diverse dropout conditions in Figure 5.

Qualitative results are illustrated in Fig. 4, which presents the 3D t-SNE visualizations of multimodal feature spaces. The figure includes the individual feature spaces of RGB, Depth, Event, and LiDAR, as well as the fused features obtained through our AnySeg framework. These visualizations provide several key insights:

**(I) Distinct Clusters in Individual Modalities:** The 3D t-SNE plots of individual modalities reveal distinct clusters corresponding to semantic classes. However, modalities like Event and LiDAR exhibit greater feature dispersion, reflecting their limited discriminative power when used independently. While RGB and Depth features display relatively better cluster separability, overlaps between certain semantic classes persist, underscoring the challenges of relying solely on individual modalities.

**(II) Improved Separability in Paired Modalities:** Feature spaces derived from modality combinations, such as RGB+Depth or RGB+Depth+Event, show marked improvements in cluster separability and compactness when processed through the AnySeg framework. This highlights the benefit of leveraging complementary information across modalities. For instance, integrating RGB and Depth reduces the ambiguities present in individual modalities, yielding more cohesive and distinct clustering.

**(III) Robust Fused Feature Space:** The fused feature space of RGB+Depth+Event+LiDAR, as modeled by the AnySeg framework, demonstrates the most compact and well-separated clusters among all configurations. This indicates the framework's effectiveness in integrating multimodal information and enhancing the semantic distinctions between classes. Compared to individual or paired modalities, the fused features provide a more robust representation of the underlying data.

**(IV) Mitigation of Ambiguities in Challenging Modalities:** The AnySeg framework effectively addresses the limitations of challenging modalities like Event and LiDAR. Through robust cross-modal fusion mechanisms, it compensates for the weaknesses of these modalities, resulting in consistent and accurate segmentation across diverse scenarios. The framework's ability to integrate complementary strengths across modalities ensures superior performance in handling complex data distributions.

These qualitative results corroborate the quantitative findings reported in Tab. 1 and Tab. 2. The clear clustering of semantic classes and improved separability in the fused feature space highlight the superiority of the proposed framework in learning robust, multimodal feature representations. Overall, Fig. 4 visually demonstrates the capability of the AnySeg framework to effectively harness multimodal information for achieving robust semantic segmentation in both individual and multimodal settings.

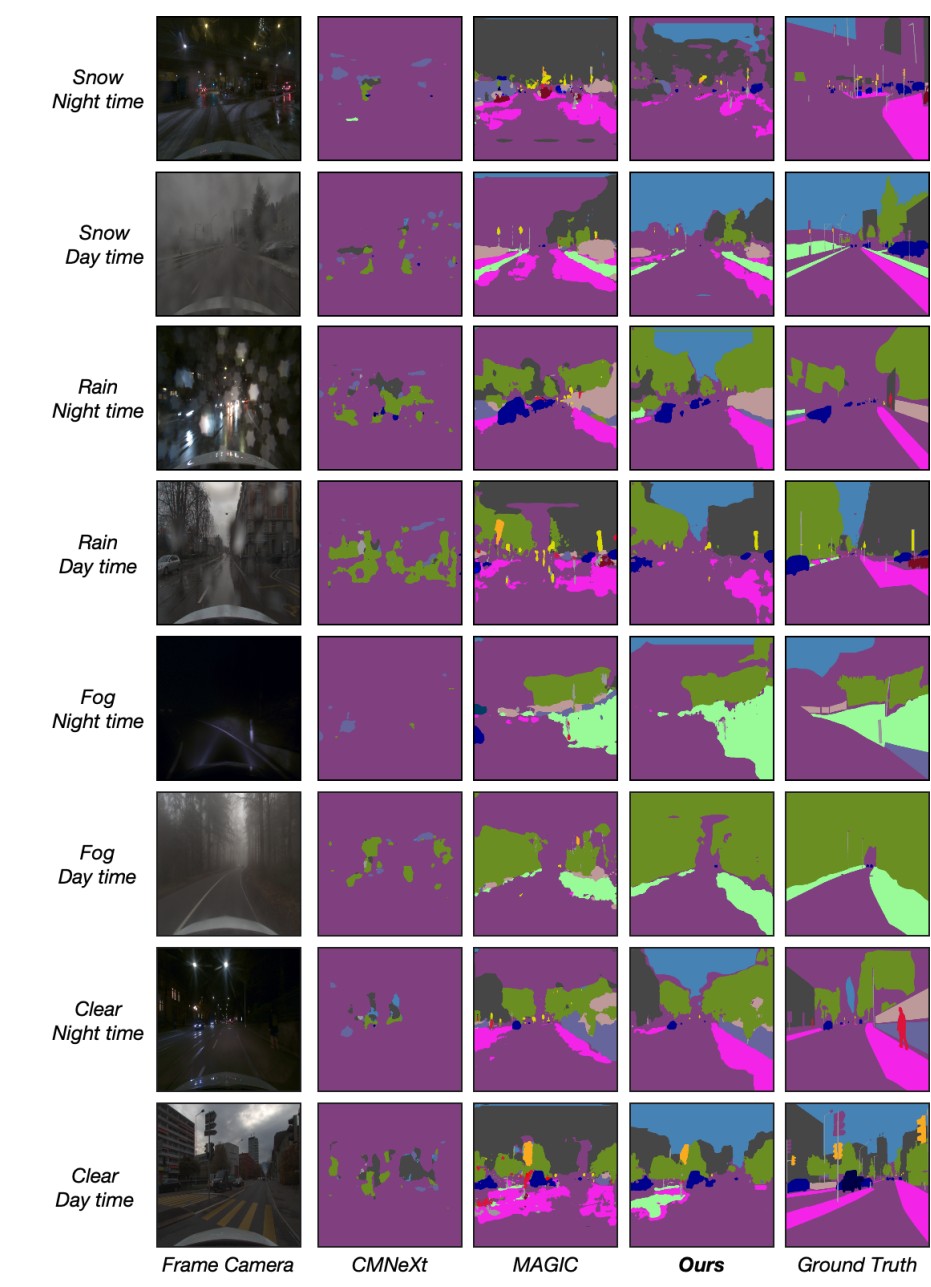

Figure 5: Additional qualitative comparisons that highlight our method's robustness under diverse dropout conditions.

## A.4 MORE EXPERIMENTAL RESULTS

The experimental results in Table 10 validate the superiority of our proposed method in modality-agnostic validation on the DELIVER dataset, utilizing three modalities. Across individual modalities, our method achieves the highest performance in RGB (47.44%) and Event (17.33%), significantly outperforming MAGIC by margins of +14.48% and +15.18%, respectively, while also maintaining competitive results in Depth (52.48%). For paired modalities, our approach shows strong cross-modal fusion capabilities, achieving substantial improvements in RGB+Event (RE, 47.65%, +14.40%) and Depth+Event (DE, 52.61%, +3.39%) over MAGIC. Although MAGIC achieves the highest performance in RGB+Depth (RD, 62.52%) and combined modalities (RDE, 62.49%), our method

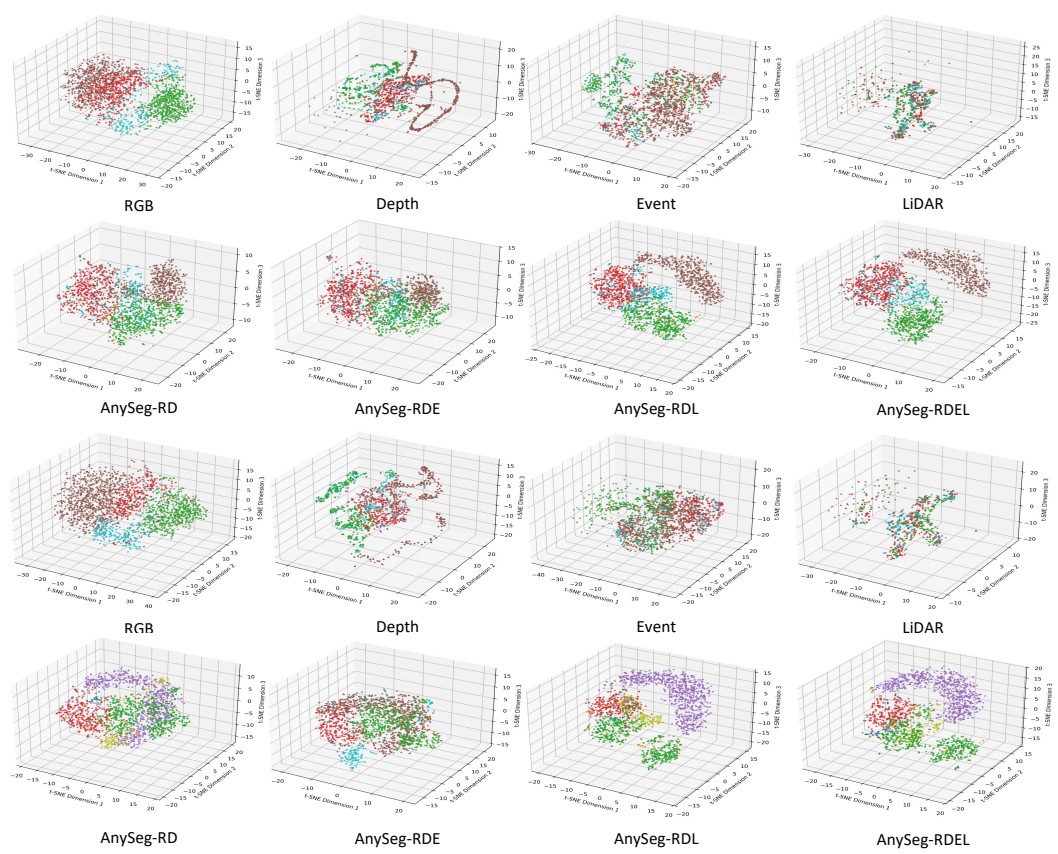

Figure 6: t-SNE visualization of multi-modal features (RGB, Depth, Event, and LiDAR) extracted by the SegFormer backbone and the features of our AnySeg framework.

Table 10: Results of modality-agnostic validation with three modalities (R: RGB, D: Depth, E: Event) on DELIVER.

| Method | Training | Anymodal Evaluation | | | | | | | Mean |
|---|---|---|---|---|---|---|---|---|---|
| | | R | D | E | RD | RE | DE | RDE | |
| CMNeXt (Zhang et al., 2023b) | RDE | 2.69 | 0.21 | 0.78 | 48.04 | 6.92 | 2.19 | 59.84 | 17.24 |
| MAGIC (Zheng et al., 2024c) | | 32.96 | **55.90** | 2.15 | **62.52** | 33.25 | **56.00** | **62.49** | 43.61 |
| Ours | | **47.44** | 52.48 | **17.33** | 61.04 | **47.65** | 52.61 | 60.62 | **48.45** |
| *w.r.t* SoTA | | +14.48 | -3.42 | +15.18 | -1.48 | +14.40 | -3.39 | -1.87 | +4.84 |

performs competitively in these settings with RDE achieving 60.62. Importantly, our method achieves the highest mean mIoU of 48.45%, surpassing MAGIC by +4.84% and CMNeXt by +31.21%, demonstrating superior generalization across diverse modality configurations. These results underscore the robustness of our framework in addressing modality-agnostic challenges, effectively leveraging complementary information across modalities and maintaining balanced performance even in the presence of weaker modalities like Event.

## A.5 TRAINING EFFICIENCY ANALYSIS

To further address concerns regarding computational cost, we provide a detailed comparison of training time, GPU memory usage, and inference memory usage across representative baseline methods. All experiments were conducted using the SegFormer-B0 backbone on 8 NVIDIA 3090 GPUs with a batch size of 2 per GPU.

Table 11: Ablation study on the effect of different parameters for $L_{mad}$ in our framework on MUSES dataset (Brödermann et al., 2024).

| $\lambda$ | F | $\Delta \uparrow$ | E | $\Delta \uparrow$ | L | $\Delta \uparrow$ | FE | $\Delta \uparrow$ | FL | $\Delta \uparrow$ | EL | $\Delta \uparrow$ | FEL | $\Delta \uparrow$ | Mean | $\Delta \uparrow$ |
|---|---|---|---|---|---|---|---|---|---|---|---|---|---|---|---|---|
| 1 | 43.97 | - | 22.33 | - | 31.90 | - | 44.82 | - | 48.61 | - | 35.14 | - | 48.33 | - | 39.30 | - |
| 10 | 43.84 | -0.13 | 23.21 | +0.88 | 32.71 | +0.81 | 44.08 | -0.74 | 49.16 | +0.55 | 34.97 | -0.17 | 48.08 | -0.25 | 39.44 | +0.14 |
| 20 | 44.08 | +0.11 | 22.76 | +0.43 | 32.35 | +0.45 | 44.37 | -0.45 | 49.33 | +0.72 | 34.73 | -0.41 | 48.79 | +0.46 | 39.49 | +0.19 |
| **50** | 43.71 | -0.26 | 23.00 | +0.67 | **34.70** | +2.80 | 44.18 | -0.64 | 49.13 | +0.52 | **37.23** | +2.09 | **48.79** | +0.46 | **40.11** | +0.81 |
| 60 | 44.02 | +0.05 | 22.74 | +0.41 | 33.82 | +1.92 | 44.29 | -0.53 | 49.36 | +0.75 | 36.69 | +1.55 | 48.54 | +0.21 | 39.92 | +0.62 |
| 80 | 43.84 | -0.13 | 22.86 | +0.53 | 33.78 | +1.88 | 44.25 | -0.57 | 49.43 | +0.82 | 36.57 | +1.43 | 48.72 | +0.39 | 39.92 | +0.62 |
| 100 | 43.75 | -0.22 | 22.87 | +0.54 | 34.00 | +2.10 | 44.17 | -0.65 | 49.36 | +0.75 | 36.60 | +1.46 | 48.64 | +0.31 | 39.91 | +0.61 |

Table 13: Ablation study on the effect of different parameters for add $L_{cmd}$ with $L_{umd}$ on MUSES dataset (Brödermann et al., 2024). w/o means the framework is only trained with $L_{mad} + L_{umd}$.

| $\beta$ | F | $\Delta \uparrow$ | E | $\Delta \uparrow$ | L | $\Delta \uparrow$ | FE | $\Delta \uparrow$ | FL | $\Delta \uparrow$ | EL | $\Delta \uparrow$ | FEL | $\Delta \uparrow$ | Mean | $\Delta \uparrow$ |
|---|---|---|---|---|---|---|---|---|---|---|---|---|---|---|---|---|
| w/o | 45.82 | - | 19.26 | - | 31.79 | - | 45.88 | - | 51.11 | - | 33.56 | - | 50.60 | - | 39.72 | - |
| 1 | 45.93 | +0.11 | 18.76 | -0.50 | 31.84 | +0.05 | 45.96 | +0.08 | 51.22 | +0.11 | 33.49 | -0.07 | 50.82 | +0.22 | 39.72 | 0.00 |
| 3 | 46.04 | +0.22 | 17.74 | -1.52 | 31.42 | -0.37 | 46.08 | +0.20 | 51.27 | +0.16 | 33.46 | -0.10 | 50.99 | +0.39 | 39.57 | -0.15 |
| 5 | 46.19 | +0.37 | 17.27 | -1.99 | 31.03 | -0.76 | 46.27 | +0.39 | 51.34 | +0.33 | 33.40 | -0.16 | 51.05 | +0.45 | 39.51 | -0.21 |
| 7 | 46.21 | +0.39 | 17.40 | -1.86 | 31.06 | -0.73 | 46.37 | +0.49 | 51.29 | +0.18 | 33.79 | +0.23 | 51.12 | -0.12 | 39.60 | -0.12 |
| **10** | 46.01 | +0.19 | 19.57 | +0.31 | 32.13 | +0.34 | 46.29 | +0.41 | 51.25 | +0.14 | 35.21 | +1.65 | 51.14 | +0.54 | 40.23 | +0.51 |
| 13 | 46.57 | +0.75 | 18.24 | -1.02 | 30.88 | -0.91 | 46.74 | +0.86 | 51.09 | -0.02 | 33.88 | +0.32 | 50.76 | +0.16 | 39.74 | +0.02 |
| 15 | 46.03 | +0.21 | 14.10 | -8.90 | 31.12 | -3.58 | 45.99 | +1.81 | 50.97 | +1.84 | 31.42 | -5.81 | 50.49 | -0.11 | 38.59 | -1.13 |
| 20 | 45.95 | +0.13 | 15.19 | -7.81 | 30.61 | -4.09 | 45.80 | +1.62 | 51.19 | +2.06 | 30.55 | -6.68 | 50.41 | +1.62 | 38.53 | -1.58 |

## A.6 MORE RESULTS IN ABLATION STUDIES

As shown in Table 12, our proposed **AnySeg** achieves competitive efficiency: training time and memory usage are comparable to CMNeXt, while inference memory usage is significantly lower. Although the training cost is moderately higher than MAGIC, this

Table 12: Comparison of training and inference efficiency across methods using SegFormer-B0.

| Method | Training Time (hrs) | Training Mem. (GB) | Inference Mem. (GB) |
|---|---|---|---|
| Any2Seg | 18.54 | 23.9 | 7.1 |
| MAGIC | 9.27 | 8.6 | 7.4 |
| CMNeXt | 13.76 | 9.8 | 8.9 |
| **AnySeg (Ours)** | 15.17 | 14.5 | **6.8** |

trade-off is justified by the substantial improvements in segmentation accuracy and robustness under missing-modality conditions, as demonstrated in our experimental results.

## A.7 COMPUTATIONAL COMPLEXITY ANALYSIS

We compare the parameter count and GFLOPs (at a 1024×1024 resolution) for our method and key baselines. As shown in Table 14, our method, **AnySeg**, achieves high segmentation performance with a significantly more efficient inference model. While training involves a more complex teacher, the overhead is limited to the training phase, with the final deployed model being the lightweight student, ensuring practical feasibility.

Table 14: Computational Complexity Comparison (at 1024×1024 resolution)

| Method | Parameters (M) | GFLOPs |
|---|---|---|
| CMNeXt | 10.30 | 44.78 |
| MAGIC | 24.74 | 444.35 |
| Any2Seg | 24.73 | 236.44 |
| **AnySeg (Ours)** | **3.72** | **33.82** |

## A.8 SEGMENTATION PERFORMANCE OF THE TEACHER NETWORK

We present the segmentation results for both the multimodal teacher and student networks, using SegFormer-B0 as the backbone. The performance is evaluated on the DELIVER dataset across

multiple modality subsets: RGB (R), Depth (D), Event (E), and LiDAR (L), and their combinations.As shown in Table 15, the teacher network (Seg-B2) achieves strong performance across all modality combinations, with particularly high results on full-modality inputs. However, the student network (Seg-B0), despite having a smaller backbone, closely approximates or even surpasses the teacher on several modality subsets, demonstrating its ability to perform well in more resource-efficient settings. Importantly, the teacher's performance relies on the full-modality input and is not optimized for inference in missing-modality scenarios, making the student network essential for anymodal deployment.

Table 15: Anymodal Segmentation Performance on DELIVER Dataset (Modalities: R = RGB, D = Depth, E = Event, L = LiDAR)

| Method | R | D | E | L | RD | RE | RL | DE | DL | EL | RDE | RDL | REL | DEL | RDEL | **Mean** |
|---|---|---|---|---|---|---|---|---|---|---|---|---|---|---|---|---|
| Teacher (Seg-B2) | 39.02 | 60.11 | 2.07 | 0.31 | 67.21 | 39.11 | 39.04 | 60.92 | 60.15 | 1.99 | 67.64 | 67.82 | 39.06 | 60.95 | 67.95 | 44.89 |
| Student (Seg-B0) | 47.11 | 52.17 | 17.33 | 19.01 | 60.37 | 47.49 | 48.13 | 52.82 | 52.29 | 21.47 | 60.16 | 60.60 | 47.98 | 52.44 | 60.26 | 46.64 |

# B   THE USE OF LARGE LANGUAGE MODELS (LLM)

We used OpenAI's GPT-4o to assist with the refinement and proofreading of certain sentences in this paper. The LLM was used exclusively to enhance the clarity and coherence of our writing. All content contributions are made by the authors.

