# OpenReview forum: "Learning Robust Anymodal Segmentor with Unimodal and Cross-modal Distillation"
_ICLR.cc/2026/Conference — ICLR 2026 Conference Withdrawn Submission_

### Official Review · Reviewer_rTLZ · 2025-10-31

**Soundness:** 3
**Presentation:** 2
**Contribution:** 2
**Rating:** 2
**Confidence:** 4

**Summary:**

This paper presents a method to tackle the missing modality problem when learning segmentation models from multiple sensor inputs. The method designs unimodal distillation and cross-modal distillation at the feature and prediction levels to improve model performance with incomplete modalities. Experiments are conducted on synthetic and real-world multi-sensor datasets for segmentation.

**Strengths:**

- The proposed method improves model performance at various missing modality scenarios compared to existing approaches.
- The ablation study on the sensitivity of hyperparameters is comprehensive.

**Weaknesses:**

- The proposed method lacks sufficient novelty. Both unimodal and cross-modal knowledge distillation have been extensively explored in previous works on missing-modality learning, e.g., [R1–R5]. The paper does not adequately review this substantial body of literature nor clearly delineate how the proposed approach introduces new insights or technical advances beyond existing methods. Furthermore, the experimental evaluation does not include comparisons against these representative baselines, making it difficult to assess the claimed contributions.

[R1] Wang Q, Zhan L, Thompson P, Zhou J. Multimodal learning with incomplete modalities by knowledge distillation. In Proceedings of the 26th ACM SIGKDD International Conference on Knowledge Discovery & Data Mining 2020 (pp. 1828-1838).
[R2] Garcia NC, Morerio P, Murino V. Modality distillation with multiple stream networks for action recognition. In Proceedings of the European Conference on Computer Vision (ECCV) 2018 (pp. 103-118).
[R3] Hu M, Maillard M, Zhang Y, Ciceri T, La Barbera G, Bloch I, Gori P. Knowledge distillation from multi-modal to mono-modal segmentation networks. In International Conference on Medical Image Computing and Computer-Assisted Intervention 2020 (pp. 772-781).
[R4] Wei S, Luo C, Luo Y. Mmanet: Margin-aware distillation and modality-aware regularization for incomplete multimodal learning. In Proceedings of the IEEE/CVF Conference on Computer Vision and Pattern Recognition 2023 (pp. 20039-20049).
[R5] Li M, Yang D, Zhao X, Wang S, Wang Y, Yang K, Sun M, Kou D, Qian Z, Zhang L. Correlation-decoupled knowledge distillation for multimodal sentiment analysis with incomplete modalities. In Proceedings of the IEEE/CVF Conference on Computer Vision and Pattern Recognition 2024 (pp. 12458-12468).

- The claimed “strong multimodal teacher” appears to be implemented by simply using multiple modality-specific branches and averaging their feature representations, which is a fairly standard design. It is unclear what novel architectural advances this approach introduces beyond existing multimodal fusion methods.

- Are there pixel-wise misalignments among the four modalities? Such misalignments could negatively affect the effectiveness of the feature averaging operation.

- The methodological description lacks some clarity. In Eq. 3, it is not explained why only modality r and modality d are considered. Moreover, the architecture of the anymodal segmentor is insufficiently described. Is it designed as a multi-stream network with a separate branch for each modality? Additional architectural details are needed to clarify the model design and its underlying assumptions.

- In Table II, the proposed method reports a score of 46.64, while the SOTA baseline is 45.05. The claimed improvement of +6.15 appears to be incorrect based on these values. Please verify the calculation and correct the reported performance gain.

**Questions:**

- The authors need to justify the specific methodological novelty beyond prior unimodal/cross-modal distillation works, e.g., [R1–R5]. What new mechanism or theoretical insight does the proposed approach introduce?

- A discussion and comparison with those closely related missing modality knowledge distillation methods are needed, highlighting key differences in terms of the teacher model architecture, the unimodal and cross-modal distillation, and justifying the advantages.

- The methodology descriptions need to be significantly improved in terms of clarity and technical contributions.

---

### Official Review · Reviewer_1FsZ · 2025-10-31

**Soundness:** 3
**Presentation:** 3
**Contribution:** 2
**Rating:** 4
**Confidence:** 5

**Summary:**

The paper introduces AnySeg, a robust framework for semantic segmentation in scenarios with missing or incomplete input modalities. The method employs a two-stage teacher-student paradigm: first, a strong multimodal teacher is trained using the proposed Parallel Multimodal Learning (PML) strategy; second, a student model—the “anymodal” segmentor—is distilled from the teacher via both unimodal and cross-modal knowledge distillation, complemented by a modality-agnostic prediction-level distillation loss. The framework is evaluated on both real-world (MUSES) and synthetic (DELIVER) multi-sensor datasets, demonstrating substantial improvements over state-of-the-art baselines, particularly in challenging cases with missing modalities. Overall, the approach is clearly presented, technically sound, and shows convincing empirical results.

**Strengths:**

1. The paper addresses the underexplored and critical issue of unimodal bias in multimodal semantic segmentation, with a precise focus on the robustness to missing modalities—a typical and practically important scenario for multi-sensor systems.

2. The paper is easy to follow.

**Weaknesses:**

1. Evaluation is limited to the SegFormer-B0 backbone. Including results with larger backbones (e.g., B3 or B5) or other architectures would help demonstrate scalability.

2. The current ablations focus mainly on hyperparameters. Analyzing robustness to noise or corruption, not just missing modalities, would better support claims of generalization.

3. The teacher–student setup effectively doubles training cost. The discussion of the trade-off between efficiency and performance is brief and could be expanded.

4. The teacher model is trained using a relatively simple PML approach, averaging features across modalities. This may limit the teacher’s performance, as it does not capture complex inter-modality interactions, complementarity, or uncertainty. Consequently, the student’s performance depends heavily on the distillation process rather than learning from a fully optimized multimodal fusion.

5. The framework relies on a weighted sum of multiple loss terms. The choice of these weights can be sensitive and may not generalize easily across different datasets or modality combinations.

**Questions:**

1. How sensitive is the performance to the architecture choice (e.g., SegFormer-B0 to SegFormer-B5 or other semantic segmentation models)?

2. How does AnySeg behave under noisy modality inputs rather than fully missing ones?

---

### Official Review · Reviewer_8owH · 2025-11-01

**Soundness:** 3
**Presentation:** 3
**Contribution:** 3
**Rating:** 4
**Confidence:** 4

**Summary:**

Existing multimodal semantic segmentation methods have made progress in handling modality-incomplete scenarios. However, unimodal bias remains unresolved—for example, Any2Seg’s mIoU drops sharply from 68.21 (RGB+Depth, RD) to 39.02 (only RGB, R) when depth input is missing. To tackle this, the paper proposes the first framework for learning robust anymodal segmentors: it handles missing modalities by distilling both unimodal and cross-modal knowledge, and introduces a Parallel Multimodal Learning (PML) strategy to build a strong teacher model, enhancing robust segmentation under incomplete inputs.

**Strengths:**

The motivation is clear and convincing.

The experiment is detailed, and the performance gain of PML is significant.

**Weaknesses:**

The cross-modal distillation is widely used in multimodal segmentation with missing modalities [1][2][3]. What is the new idea proposed in PML?

Besides, the imbalanced learning in multimodal learning with the missing modalities is also studied by existing methods[1]. It introduces extra unimodal regularizers. The author should compare the proposed unimodal distillation with it.

Comparasion with recent SOTA methods, such as works [3][4]

The method includes three hyperparameters and performs individual parameter ablation. How then should the coupling effect between these parameters be evaluated? How should an optimal set of parameter combinations be determined?








[1] Mmanet: Margin-aware distillation and modality-aware regularization for incomplete multimodal learning, CVPR2023

[2] Learning Modality-agnostic Representation for Semantic Segmentation from Any Modalities ECCV2024

[3] OGP-Net: Optical Guidance Meets Pixel-Level Contrastive Distillation for Robust Multi-Modal and Missing Modality Segmentation, AAAI2025

[4] Benchmarking multi-modal semantic segmentation under sensor failures: Missing and noisy modality robustness CVPRW2025

**Questions:**

see the weakness

---

### Official Review · Reviewer_PDeV · 2025-11-01

**Soundness:** 2
**Presentation:** 3
**Contribution:** 2
**Rating:** 4
**Confidence:** 4

**Summary:**

This paper introduces an anymodal semantic segmentation framework aimed at tackling the challenge of missing modalities during inference in multimodal scenarios. The framework comprises a multimodal teacher model and an anymodal student model. The teacher is trained using a parallel multimodal learning strategy, simultaneously processing all available modalities. The student model then learns from the teacher through a comprehensive distillation approach. Experimental results show that the proposed framework delivers strong performance across diverse modality combinations and exhibits particularly robust behavior under conditions where some modalities are absent.

**Strengths:**

- Tackles a practically important and underexplored problem: robustness to missing modalities in multimodal segmentation, relevant to autonomous driving, robotics, and remote sensing.
- Clear and well-justified motivation; the proposed method is well-designed and directly addresses the stated problem.
- Empirical results demonstrate substantial improvements, validating the effectiveness of the proposed algorithm.
- Well-written and easy to follow.

**Weaknesses:**

- Why the teacher model calculates the mean of multimodal features? The importance and semantic contributions of each modality to the segmentation task might vary.
- Why the $L_{umd}$ introduces performance loss in Table 3?

**Questions:**

See weaknesses.

---

### Note · Authors · 2025-12-09

**Comment:**

Dear Area Chair and Reviewers,

I would like to express my sincere gratitude for the time and effort you have dedicated to reviewing my paper. I greatly appreciate your thoughtful comments, constructive feedback, and valuable suggestions. Your insights have significantly contributed to improving the quality of the paper, and I am truly grateful for your support.

After careful consideration, I have decided to withdraw my submission. I hope to address the points raised by the reviewers and make further improvements to the work in the future.

Once again, thank you for your time and for providing such constructive feedback.

Best regards,
Authors

**Withdrawal Confirmation:**

I have read and agree with the venue's withdrawal policy on behalf of myself and my co-authors.